# Role of Cav1.3 Channels in Brain–Heart Interactions: An Unexpected Journey

**DOI:** 10.3390/biomedicines13061376

**Published:** 2025-06-04

**Authors:** Jean-Baptiste Reisqs, Yvonne Sleiman, Michael Cupelli, Mohamed Boutjdir

**Affiliations:** 1Cardiovascular Research Program, VA New York Harbor Healthcare System, New York, NY 11209, USA; jeanbaptiste.reisqs@gmail.com (J.-B.R.); yvonne_sleiman@hotmail.com (Y.S.); mcupelli@gmail.com (M.C.); 2Department of Medicine, Cell Biology and Pharmacology, State University of New York Downstate Health Sciences University, New York, NY 11203, USA; 3Department of Medicine, New York University Grossman School of Medicine, New York, NY 10016, USA

**Keywords:** Cav1.3, brain, heart, neuron, cardiomyocyte, electrophysiology

## Abstract

The intricate brain–heart interaction, essential for physiological balance, is largely governed by the autonomic nervous system (ANS). This bidirectional communication, involving both the sympathetic and parasympathetic branches of the ANS, is critical for maintaining cardiac homeostasis. Dysregulation of the ANS is a significant factor in cardiovascular diseases. Beyond the ANS, higher brain functions, particularly through interoceptive prediction, contribute to this dynamic interplay. The Cav1.3 L-type calcium channel, expressed in both the central nervous system (CNS) and the heart, is crucial for this interaction. Cav1.3, a key regulator of cellular excitability, exhibits genetic variations that are linked to both neurological and cardiac disorders, highlighting its pivotal role in the brain–heart axis. This review aims to delve into the under-explored role of Cav1.3 in brain–heart interaction, specifically examining how it modulates ANS activity and, consequently, the cardiac function. This will illuminate its significant role in the broader context of brain–heart interactions.

## 1. Introduction

The heart undergoes extensive neural innervation during development, which establishes a vital brain–heart interaction. This homeostatic relationship between the brain and the heart is essential for maintaining normal physiological function. This interaction is primarily regulated by the autonomic nervous system (ANS). Notably, aberrant ANS activity plays a critical role in the pathophysiology of cardiovascular diseases [1]. The ANS branches include the sympathetic nervous system (SNS) and the parasympathetic nervous system (PNS) (Figure 1A). The SNS plays a pivotal role in the body’s ‘fight or flight’ response and releases neurotransmitters, with noradrenaline (NA) being the primary messenger, across the neurocardiac junction [2]. Cardiac sympathetic innervation abnormalities contribute to various heart diseases, such as long QT syndrome or catecholaminergic polymorphic ventricular tachycardia (CPVT), and have been directly linked to their pathogenesis [3,4]. Notably, sympathetic hyperinnervation was the initial form of neural remodeling identified as being connected to the development of arrhythmias in humans [5]. Regions of sympathetic hyperinnervation are characterized by a higher density of nerve fibers, resulting in an overabundance of norepinephrine and consequently leading to tachycardia (Figure 1B) [6]. On the other hand, SNS denervation also plays a role in heart rhythm instability. This process can cause bradycardia in patients and an increase in repolarization dispersion, making it arrhythmogenic (Figure 1C) [7].The PNS promotes relaxation, often referred to as the ‘rest and digest’ state. By releasing acetylcholine (Ach), it counteracts the effects of the SNS, maintaining heart function homeostasis [8]. Aberrant activity of the PNS was also implicated in heart diseases like heart failure [9,10]. Heart rate variability (HRV) is a common method for studying the influence of ANS on cardiac function [10,11]. The higher brain can also influence ANS activity and heart function. In fact, human neuroimaging shows that cardiovascular sensation involves not just the cortex, but a broader brain network, emphasizing the brain’s predictive role in interoception. This concept, supported by constructivist theories, suggests that perceptions of bodily states are influenced by the brain’s predictions, with ‘interoceptive prediction errors’ triggering compensatory adjustments [12]. A revised neurovisceral integration model highlights a hierarchical network from cortex to heart, allowing for nuanced autonomic responses [13]. The cortex especially plays a crucial role in modulating the limbic system, acting as a top-down control mechanism to regulate emotions, memories, and behaviors [14,15]. The cortex interacts with limbic structures through various neurotransmitter such as glutamate. The cortex allow an appropriate and adaptative behavioral responses to various situations [16]. This multi-level regulation integrates visceral signals and adapts to internal and external demands. The inherent dynamism and integrated nature of heart–brain interactions suggest that interventions designed to enhance autonomic flexibility, such as biofeedback or stress management techniques, hold potential benefits for both neurological and cardiac function [1].

The central nervous system (CNS), like the heart, is made up of excitable cells, which depend on ion channels for proper functioning. Among these ion channels are three fundamental types: sodium, potassium, and calcium channels. L-type calcium channels (LTCCs) in the CNS exist as two subunit forms, Cav1.2 and Cav1.3, which are involved in short- and long-term plasticity [17]. Their differing biophysical properties, activation kinetics, and localization within neurons contribute to their distinct roles. While Cav1.2 is more involved in synaptic plasticity, dendritic integration, and activity-dependent gene expression, Cav1.3 plays a greater role in spontaneous neuronal activity, neurotransmitter release, and neuronal survival [17,18]. Cav1.3 is a macromolecular complex consisting of a central ion conducting protein (α1D subunit) and additional accessory channel subunits (α2γ and β), which modulate the trafficking and gating properties of Cav1.3 [19]. The α1D subunit of Cav1.3, coded by the *CACNA1D* gene, is a transmembrane protein composed of four homologous repeats (DI-DIV), each containing six transmembrane segments (Figure 2). These segments house the voltage sensors and pore loops, which together construct the functional ion channel [20]. Interestingly, genetic variants of Cav1.3 cause congenital deafness, epilepsy, and autism spectrum disorder in the CNS and sinus bradycardia, heart failure, and atrioventricular block in the heart [21,22,23,24].

The purpose of this review is to shed light on the relatively unexplored role of the Cav1.3 channel in the brain–heart interaction. We systematically describe the expression and function of Cav1.3 within the CNS and how this ion channel could impact the activity of the ANS and the regulation of cardiac function.

## 2. Neuronal and Cardiomyocyte Action Potential

Neuronal and cardiomyocyte action potentials (APs) exhibit distinct electrophysiological characteristics, correlated with their respective functions. Neuronal and cardiac APs feature depolarization, caused by the entry of sodium ions, and repolarization, caused by the exit of potassium ions [25]. Neuronal APs are characterized by rapid kinetics, optimized for the fast propagation of nerve impulses, whereas cardiac APs, of longer duration, enable excitation–contraction coupling and the synchronization of myocardial contraction [25,26]. Specifically, the human neuronal AP duration is around 1–2 ms, whereas the human cardiac AP duration is around 200–300 ms (Figure 3). The shape and duration of cardiac APs vary significantly across different regions of the heart. Ventricular APs exhibit a longer duration compared to atrial APs, primarily due to an extended plateau phase. Additionally, the sinoatrial node (SAN) cells display a distinctly different AP shape with a spontaneous depolarization phase [25]. A key distinction between neuronal and cardiac APs is the absence of a plateau phase in neurons. This is due to the limited influence of calcium channels on the neuronal AP waveform. Neurons also possess a short refractory period, facilitating rapid firing. In terms of propagation, neuronal signals travel along axons and across synapses, while cardiac signals spread through gap junctions between cells (Figure 3) [27,28].

LTCCs, Cav1.2 and Cav1.3, play distinct roles in different regions of the heart. More specifically in the atria and ventricles, they are essential for excitation–contraction coupling, while in the SAN, the Cav1.3 channels are crucial for pacemaker function [29,30]. Like in the heart, Cav1.2 and Cav1.3 are present in the brain. LTCCs in the CNS influence postsynaptic activity through their voltage-dependent gating of calcium influx during significant membrane potential depolarizations and are not primarily responsible for neurotransmitter release [31]. LTCCs are significant regulators of neuronal excitability, primarily by enabling burst APs. They also regulate after-hyperpolarization, thereby influencing burst frequency and contributing to the fine-tuning of neuronal excitability [32,33].

Through their dual actions in the heart and brain, LTCCs emerge as pivotal regulators of interorgan communication. Notably, the Cav1.3 subtype has been implicated in both neuronal and cardiac pathologies, which we will explore further in this review.

## 3. The Cav1.3 Channel in the Central Nervous System

As previously mentioned, the Cav1.3 channel is expressed in several tissues, including the brain [34]. Some of the relevant variants of this channel have been associated with multiple functional deficits in the ANS development and function, as summarized in Table 1 [35,36,37]. Cav1.3 regulates neuronal excitability, synaptic plasticity, and neuronal gene transcription. This channel contributes to various brain functions, such as emotional and drug-use behaviors and memory [18]. It is expressed in different parts of the CNS that we discuss below [38,39,40].

Firstly, Cav1.3 has been previously reported to be expressed in the hippocampus [43]. The hippocampus serves a critical function in memory, navigation, and cognition [44,45]. It is important to note that adult hippocampal neurogenesis is conserved in the brains of most mammals, including humans. This neurogenesis, in the dentate gyrus (DG) of the hippocampus, has been implicated in cognitive processes such as learning, memory, pattern separation, and cognitive flexibility [44]. One study using Cav1.3^−/−^ mice at different and specific stages of neuronal differentiation described a crucial role of Cav1.3 in neurogenesis functions in the hippocampus [46]. They observed a decrease in Cav1.3 expression associated with a decreased rate of adult hippocampal neurogenesis in the DG. These mice exhibited a significantly impaired object location discrimination during the object location memory test, suggesting a strong role of Cav1.3 in the spatial memory function of the adult brain. Another study also reported a role of Cav1.3 in dorsal hippocampal neurons during the development and survival of newborn neurons [47]. The Cav1.3^−/−^ mice exhibited a reduction in newborn neuron survival rate compared to wild-type mice, which was accompanied by a deficit in hippocampal-dependent learning and memory tasks. These two studies shed light on the role of the Cav1.3 channel in memory and cognition tasks; however, the mechanism by which Cav1.3 contributes to the survival of newborn neurons during neurogenesis remains to be studied. A recent study used Cav1.3 RNA editing in mice to increase the calcium influx into neurons [48]. In this study, genetic approaches were used to specifically target and eliminate RNA editing of the Cav1.3 channel, thereby ensuring the exclusive expression of the unedited Cav1.3 variant. In more detail, they ablated the first site of the Cav1.3 IQ-domain IQDY (ATA→ATG; I→M) residue in the mouse to slow the calcium-dependent inactivation and allow a larger entry of calcium [49]. This RNA editing of its IQ domain is restricted to neurons of the CNS. This study provided evidence that this editing recoding has functional, physiological, and behavioral importance. They showed that the loss of Cav1.3 IQ-domain RNA editing is a gain of function, increased Cav1.3 channel current, and led to enhanced neuronal excitability compared to Cav1.3 with the IQ-domain RNA editing. This larger calcium influx has the consequence of increasing neuronal excitability and enhancing hippocampal plasticity, which leads to strengthened late long-term potentiation and improved spatial learning and memory [48]. These findings contribute to elucidating how a loss of function of Cav1.3 impacts neurogenesis and differentiation.

The Cav1.3 channel is also highly expressed in the striatum. This cerebral region is involved in modulating movement and cognitive functions. Some studies report a functional role of this channel, especially in Parkinson’s disease [41,50]. Although levodopa (L-Dopa), which is a precursor to dopamine, is effective for Parkinson’s disease, it causes dyskinesia. Two preliminary studies in rat models have demonstrated that antagonizing Cav1.3 channels could offer a potential strategy for avoiding this complication [50,51]. Furthermore, a human variant of Cav1.3 (A749G) was demonstrated to induce aberrant excitability of neurons in the striatum [41]. These studies confirm the functional role of the Cav1.3 channel in the striatum and its link to Parkinson’s disease [35].

Cav1.3 variants have also been associated with autism spectrum disorder. In 2015, the A749G and G407R variants of *CACNA1D*, which cause a gain of function in channel activity, were identified as significant risk factors for autism [42]. Another gain-of-function *CACNA1D* variant (V401L) was also linked to autism spectrum disorder and epilepsy [37]. More recently, the G1169D variant was also associated with a gain of function resulting in severe neurologic symptoms [36]. Two boys carrying this variant were classified in the autism spectrum. Finally, the variant R930H causes a complex disturbance in ion channel activity. Specifically, this variant results in a loss of function in the long Cav1.3, linked to sinus node dysfunction, and also a loss of function in the brain, leading to epilepsy [22]. These variants, through their effect on channel gating, reveal the prominent and multifaceted function of the Cav1.3 LTCC in controlling cellular excitability across a range of tissues. This statement implies that Cav1.3 plays a role in brain–heart interactions. The localization and role of Cav1.3 are summarized in Table 2.

### Heart–Brain Interactions

The ANS plays a key role in regulating heart function, adjusting both the strength and speed of contractions in response to constantly changing internal and external factors [61].

The ANS includes two branches (Figure 1). The SNS is responsible for rapidly and consistently enhancing heart performance during the fight-or-flight response. The SNS stimulates the release of adrenaline and noradrenaline, which increase heart rate and blood pressure [62]. The PNS releases Ach, which slows heart rate and lowers blood pressure. It provides the main source of inhibitory input to the heart, facilitating the deceleration of heart rate [63]. By providing opposing influences, these two systems balance cardiac function and stabilize cardiac homeostasis. The ANS’s influence on the heart can be assessed through the electrocardiogram (ECG) recordings, specifically by measuring the HRV [64]. The efferent nerves of the PNS, particularly the X cranial nerve (vagus nerve), originate in the brainstem and form synapses with parasympathetic ganglia within the heart, known as intracardiac ganglia [65]. The short postganglionic nerves then extend to the SAN and the atrioventricular (AV) node, releasing Ach, which decreases heart rate and contractility. The SNS is more complex. Its origin is located in the spinal cord. Nerve fibers travel to the sympathetic ganglia located in the cervical and stellate ganglia. From there, efferent nerves proceed to the heart and connect to cardiomyocytes located in the SAN, the AV node, and the ventricles [66].

However, emotional and cognitive factors, involving higher brain centers like the limbic system and prefrontal cortex, can modulate ANS activity and its ultimate effect on the heart [67,68]. This statement implies a more complex neuronal circuit that lays the physical connection between the two organs. Specifically, this neuronal network encompasses the prefrontal cortex, the amygdala, the hippocampus, the hypothalamus, and the nuclei of the parabrachial complex at the junction between the pons and brain stem (Figure 4) [69,70,71]. In fact, the central nucleus of the amygdala, which receives projections from the prefrontal cortex and is connected to the hypothalamus and brainstem, plays a role in cardiac function control and appears to modulate the impact of emotional stimuli on the heart [72]. The hypothalamus receives information from the cortex. Research in animal models has shown that sympathectomy and vagotomy can prevent the cardiac effects of stimulating the lateral and anterior hypothalamus. Consequently, this indicates that the hypothalamus could play a role in the genesis of certain cardiac arrhythmias [73].

This review aims to elucidate the role of the Cav1.3 channel in the neuronal electrophysiology of these regions, thereby contributing to a better understanding of the origins of certain cardiac arrhythmias. The role of Cav1.3 was previously described for the neuronal excitability in the hippocampus, which led to a poor cognition task. A noteworthy study conducted on rats revealed that the hippocampus is involved in the modulation of both the SNS and PNS [74]. The researchers observed a reduction in heart rate upon activation of hippocampal neurons, providing evidence for the connection between the hippocampus and heart function, where the Cav1.3 channel could play a role.

The hypothalamus, another brain area in this limbic circuit, plays a role in ANS interactions. In particular, the paraventricular nucleus (PVN) of the hypothalamus is recognized for its influence on ANS function and has emerged as one of the most important autonomic control centers in the brain [75]. In fact, one study in rats showed that microinjection of a GABA agonist (muscimol) into the PVN produces a depressor and sympathoinhibitor response [76]. More recently, a rat study indicated that endocannabinoid neurotransmission within the PVN either increases or decreases blood pressure and heart rate [77]. The cardiovascular response is contingent upon the type of neuron modulated, either glutamatergic or GABAergic. Interestingly, two studies on animals showed a localization of Cav1.3 in the PVN [52,78]. However, the function of this channel in PVN neuron activity is not yet fully understood. Although it likely plays a role in calcium influx, the precise contribution to neuronal firing patterns is still uncertain. Controlling fear and anxiety, the ANS, which is controlled by the limbic circuit, which includes the hypothalamus, receives most of its neuronal inputs (afferent pathways) from various limbic structures, particularly the amygdala.

The amygdala, a collection of heterogeneous nuclei located in the medial temporal lobe, plays a critical role in multimodal information processing essential for emotional recognition and behavior [53,54,79]. It is noteworthy that the Cav1.3 channel is present in this structure and has a fundamental role in its neuronal activity [80]. One study conducted on Cav1.3^−/−^ mice demonstrated the importance of Cav1.3 for long-term potentiation within the amygdala, accompanied by neuronal hyperexcitability [38]. As a result, these mice showed impaired consolidation of contextually conditioned fear, proving the role of this ion channel in the neuronal activity of this brain structure. While the role of the amygdala in the ANS and the control of cardiac function is known, the role of the Cav1.3 channel in this structure remains unexplored [81].

At the junction between the pons and brain stem, the parabranchial nucleus, which expresses the Cav1.3 channel, is known to control ANS activity [82,83]. The parabranchial nucleus is a relay of sensory information to forebrain structures, including the hypothalamus and amygdala [55]. Lastly, the brainstem, another structure exhibiting Cav1.3 expression, is recognized for its role in influencing the ANS. The brainstem, with the medulla oblongata being a key component, contains central autonomic control nuclei. These structures integrate input from various brain and spinal cord areas before sending signals to the ANS to regulate cardiac function [84]. Nevertheless, the exact nature of this influence continues to be a subject of debate [85]. The Cav1.3 channel is expressed in the brainstem and contributes significantly to auditory function [56,57,86]. It is of interest that studies in mice and humans have associated Cav1.3 loss of function with both cardiac and auditory impairments [21,87]. Nonetheless, the connection between cardiac functional abnormalities and the role of Cav1.3 within the brainstem remains unexplored.

## 4. The Cav1.3 Channel in Heart

The intrinsic cardiac nervous system, comprising intracardiac neurons (ICNs), is the ultimate pathway for autonomic nerve control of the heart, facilitating sympathetic and parasympathetic communication within the cardiac tissue [88]. These ICNs are organized as interconnected networks forming ganglionated plexuses [89]. ICNs display a wide range of phenotypes, defined by the expression of cholinergic markers and, in some cases, catecholaminergic (tyrosine hydroxylase-positive) markers, as well as other substances like calbindin, neuropeptide Y, neuronal nitric oxide synthase [90]. Electrophysiological studies have further identified distinct ICN subtypes based on their electrical characteristics. This complex of ICNs is critical for regulating normal heart function and is increasingly recognized for its contribution to cardiac pathologies, especially arrhythmias. For example, heightened ICN activity is linked to atrial fibrillation [91]. The ICNs also influence susceptibility to ventricular arrhythmias by modulating how long the heart muscle remains unresponsive after a beat [92]. In heart failure, structural changes and altered electrical properties of ICNs, often involving changes in ion channel expression, elevate the risk of ventricular arrhythmias [93]. Thus, ICNs are a key cardiac cell type with a central role in both cardiac health and disease. Investigating ion channels within this system under pathological conditions is therefore of paramount importance. Investigations into the characteristics of L-type calcium channels (LTCCs) in rat intracardiac neurons (ICNs), employing pharmacological approaches and transcript analysis, have demonstrated the expression of both Cav1.2 and Cav1.3 [94,95]. Considering their established importance in neuronal contexts, notably in the action potential upstroke, the modulation of neuronal excitability, and neurotransmitter release, it is plausible that LTCCs play a key role within ICNs, underscoring the need for further research in this area [96].

From the heart aspects, for the role of Cav1.3 in brain–heart interaction, we discuss Cav1.3 function within the heart itself. In the heart and brain, Cav1.2 is the predominant L-type Ca^2+^ channel, while Cav1.3 is generally less abundant [17]. The expression and localization of Cav1.3 are developmentally regulated. Specifically, two forms of Cav1.3 have been identified: a full-length 250 kD protein, which is the predominant form prenatally, and a shorter 190 kD form. During fetal and neonatal development, the Cav1.3 protein is found in both the atria and ventricles, but it disappears from the ventricles in adults. Conversely, the shorter 190 kD form of Cav1.3 is expressed only in adult atria [97]. Moreover, Cav1.3 undergoes significant alternative splicing in the CNS and heart, a characteristic facilitated by its long C-terminal domain. For example, the full-length Cav1.3 isoform (Cav1.3_42L_) contains all regulatory domains. In contrast, two shorter splice variants, Cav1.34_2A_ and Cav1.3_43S_, lack the distal C-terminal regulatory domain or both the proximal and distal domains [98]. Alternative splicing in the Cav1.3 C-terminus is known to alter its electrophysiological properties [99]. For instance, Cav1.3_42A_ channels exhibit calcium current activation at more negative voltages and faster inactivation due to enhanced Ca^2+^-dependent inactivation [100]. Moreover, the C-terminal modulator domain in Cav1.342 isoforms can compete with calmodulin (CaM) for binding to the IQ domain [101].The ANS innervates the heart in various compartments.

First, the SNS releases NA from postganglionic fibers. This NA binds to β1-adrenergic receptors expressed on the SAN, AV node, and ventricular cardiomyocytes (Figure 4) [102]. The release of catecholamines triggers a signaling cascade that activates adenylate cyclase, leading to an increase in intracellular cAMP within cardiomyocytes. Subsequently, cAMP activates protein kinase A (PKA), a protein that phosphorylates and promotes the opening of LTCCs. This process facilitates calcium influx into the cytosol, resulting in an elevated intracellular calcium concentration and the induction of calcium-induced calcium release from the endoplasmic reticulum [103,104,105]. This activation by the SNS has a chronotropic and ionotropic effect by increasing the heart rate and cardiomyocyte contractility. In contrast, the PNS has an opposite effect on the heart function. The afferent parasympathetic nerves release Ach and bind to the M2 muscarinic receptors in SAN and atria [106]. The Cav1.3 ion channel is essential for mediating the chronotropic responses of SAN cells from the SNS activation [58]. We previously demonstrated a critical role for both PKA and protein kinase C (PKC) in modulating the Cav1.3 channel and, consequently, heart rhythm [24,60,107,108,109]. PKA-mediated phosphorylation of Cav1.3 at serine residues 1743 and 1816 led to an upregulation of channel activity in both the SAN and atria [60,107]. In contrast, PKC activation resulted in an inhibition of the channel, associated with a 50% reduction in Cav1.3 current, particularly when phosphorylation occurred at serine 81 [108,109]. A study using Cav1.2^DHP−/−^ mice, with abolished sensitivity to dihydropyridine, indicated that the Cav1.3 channel plays a critical role in mediating the adrenergic response [59]. The study suggests that selective Cav1.3 inhibitors could be used to reduce heart rate during inappropriate sinus tachycardia. Cav1.3 expression in the adult heart is limited to the atria, SAN, and AV node, where Cav1.3 functions in AV electrical conduction (Table 2) [24]. The disruption of this channel’s function can lead to an atrioventricular block of varying severity [110]. Clinically, Cav1.3 is also significant in the context of sinoatrial dysfunction and deafness syndrome (SANDD) [21,23]. Specifically, *CACNA1D* variants were associated with SA node dysfunction and deafness in Pakistani families and also a Turkish family [22,23].

In general, while Cav1.3 is typically less abundant than Cav1.2, its significance in cardiac development and its involvement in cardiac pathologies are being recognized and gaining more interest. In SAN cells, Cav1.2 and Cav1.3 have different roles due to their distinct electrophysiological properties. Cav1.2 has a high activation threshold at −30 mV with a peak activation around 0 mV. Therefore, it does not directly participate in diastolic depolarization, compared to Cav1.3 which harbors a slower inactivation kinetics and is activated at more hyperpolarized potentials [111]. However, Cav1.2 is responsible for the upstroke of the SAN AP, which is a predominantly calcium-based AP. For these reasons, Cav1.3 is more important for the SAN cells’ spontaneous activity and responses to the ANS. Its primary roles are recognized in atrial fibrillation, SAN dysfunction, and heart failure [112,113,114].

## 5. Neuromuscular Experimental Modalities and Developments

This review has shown that Cav1.3^−/−^ mouse models have been instrumental in determining the brain regions that influence cardiac function and the distribution of the Cav1.3 channel. Nevertheless, a gap exists in the literature regarding the assessment of cardiac function in these models, which warrants further investigation. Given the physiological disparities between mice and humans, which can result in inaccurate interpretations, human induced pluripotent stem cells (hiPSCs) offer a valuable alternative for studying Cav1.3’s impact on neuronal excitability and its interaction with the heart. In response to the growing need for human-relevant models, studies of hiPSC technology have surged over the past decade [115]. A key benefit of hiPSCs is their unlimited self-renewal capability, which offers an inexhaustible source of human cells and allows for the conservation of a patient’s specific genetic variants [116]. This pluripotency allows for differentiation into any human cell type, granting access to diverse human tissues. Specifically, hiPSCs can be differentiated into cardiomyocytes and/or neurons. The differentiation of cardiomyocytes into atrial, ventricular, and pacemaker-like cardiomyocytes is now established [117,118]. The emerging neuronal differentiation is now very promising. The first significant advancements in establishing in vitro neurocardiac models came from Oh and collaborators, who successfully generated NA-secreting human embryonic-stem-cell-derived sympathetic neurons (SNs) that exhibited functional coupling with neonatal rat ventricular cardiomyocytes (CMs) [119]. Building on this work, another study developed functional co-cultures using hiPSC-derived SNs and hiPSC-CMs. Their research demonstrated that hiPSC-SNs could effectively modulate the beating rate of hiPSC-CMs and promote SN maturation and development [120]. Takayama and collaborators engineered sympathetic-like and parasympathetic-like neurons capable of antagonistically regulating the hiPSC-CM beating rate, thus recapitulating the ANS regulation of heart rate [121]. However, these neurons did not secrete detectable levels of NA or Ach. Since then, a variety of studies investigating the neurocardiac connection using a microfluidic system to separate the neuronal and cardiomyocyte compartment have been published [122,123,124].

While neurocardiac studies are few, there have been key developments in neuromuscular studies that show promise for translation to cardiomyocytes. A 2D bioreactor model of the neuromuscular junction demonstrated that electrical stimulation of co-cultured motoneurons and myocytes resulted in improved maturation and function of both cell types [125]. Specifically, the stimulated co-cultured cells exhibited improved morphological development, upregulated neuronal and muscular gene expression, aggregation of Ach receptors in the vicinity of motoneurons, and responsiveness to glutamate stimulation (ensuring muscle contraction is caused solely by neuronal activation) [125]. Bioreactors offer the ability to fine-tune culture parameters to reproduce tissue-specific physiological environments in vitro, which allows for improved microenvironment simulation of disease states [126]. Environmental control can be further enhanced with microfabrication techniques, such as photopolymerization, to create polyacrylamide hydrogels with bands of alternating rigidity, mimicking the mechanical characteristics of neuromuscular junctions and promoting the clustering of Ach receptors [127]. If complete separation of the neural and muscular portions of the co-culture is desired, compartments can be divided with microchannels that are only permissive to axons [128]. Not only does compartmentalization confine the neuromuscular junctions to a prescribed region of interest, facilitating study, but both the media and any pharmacological agents can be selectively introduced to one or both cell types [128]. Many of the advancements in 2D co-culture have also been replicated in 3D co-culture, including the incorporation of microfluidic devices [129,130]. For example, Visone et al. established a 3D bioreactor for cardiac scaffolds, combining biphasic stimulation and interstitial fluid flow, that resulted in enhanced maturation, beating properties, and cardiac protein expression [131].

By generating a new hiPSC cell line with a de novo Cav1.3 variant, we could gain deeper insights into the role of this channel in neuronal activity and its influence on neurocardiac connections [132]. However, limitations such as maturity and differentiation efficiency must be considered to fully use this technology.

## 6. Conclusions

We have reviewed the established and prospective roles of the Cav1.3 channel in brain–heart interaction. This channel’s expression in key brain regions controlling ANS activity, including the hypothalamus, hippocampus, amygdala, pons, and brainstem, underscores its importance in regulating cardiac function. Furthermore, its localization and function within the SAN establish Cav1.3 as a pivotal regulator of cardiac ANS activity.

The association of Cav1.3 variants with brain and heart pathologies, such as SDDN and autism spectrum disorder with sinus bradycardia, highlights the need for further investigation. Employing novel technologies and methods to explore Cav1.3 function offers a promising avenue to deepen our understanding of this critical calcium channel’s role in brain–heart interactions.

## Figures and Tables

**Figure 1 biomedicines-13-01376-f001:**
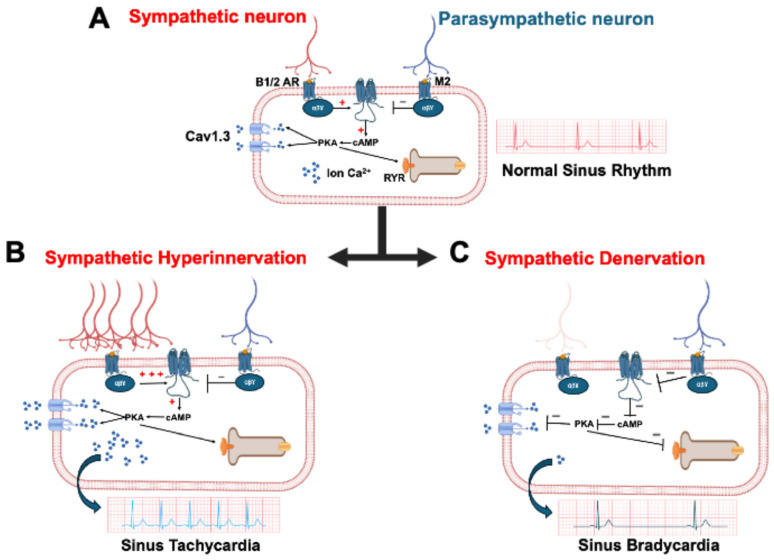
**Autonomous regulation of heart rate by sympathetic and parasympathetic innervation.** (**A**) balance between sympathetic (red, β1/2 AR, increased cAMP/PKA, calcium release) and parasympathetic (blue, M2 receptors, inhibited cAMP/PKA) nerve activity regulates cardiomyocyte electrical activity and heart rate. In (**B**), sympathetic hyperinnervation leads to the overactivation of β-adrenergic receptors and a strong increase in cAMP and PKA activity, which promotes excessive intracellular calcium release and increased cardiac electrical activity, leading to tachycardia. As shown in (**C**), sympathetic denervation reduces the activation of β-adrenergic receptors and thus the cAMP/PKA signaling, which decreases calcium release. This results in slowed or abnormal cardiac electrical activity, represented by bradycardia on the ECG. In red the sympathetic pathway and blue the parasympathetic pathway.

**Figure 2 biomedicines-13-01376-f002:**
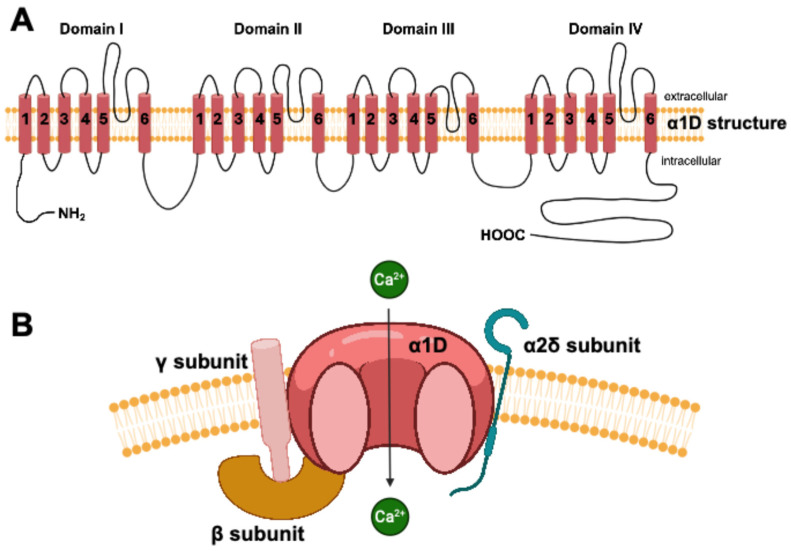
**Structural organization of the L-type calcium channel Cav1.3.** (**A**) The Cav1.3 channel comprises four transmembrane domains (I–IV), each with six segments (S1–S6). S4 segments are crucial for voltage sensing, while the loop between S5 and S6 forms the ion-conducting pore. Both the N-terminal and C-terminal ends are intracellular, with the C-terminus being involved in the functional regulation of the channel. (**B**) Membrane assembly of the functional channel, including the pore-forming α1D subunit and the regulatory β, γ, and α2δ subunits, illustrating the passage of calcium ions.

**Figure 3 biomedicines-13-01376-f003:**
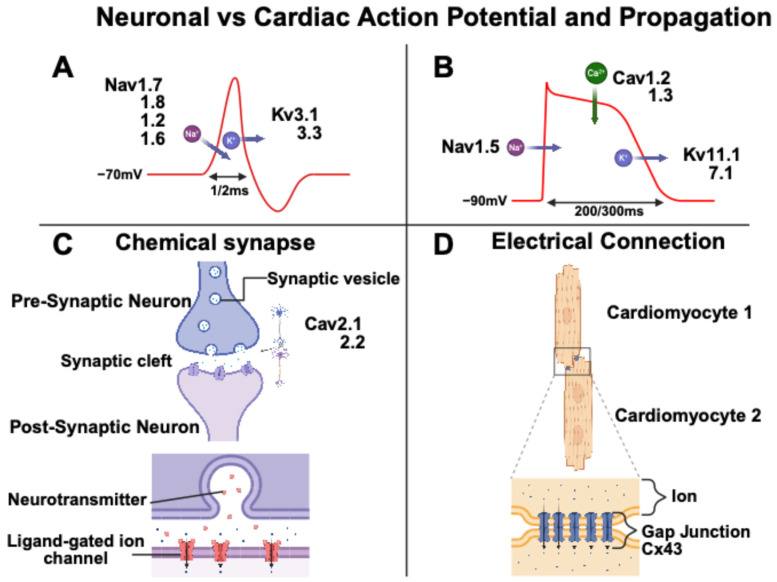
**Action potential propagation and signal transmission in neurons and cardiomyocytes.** (**A**) Representation of a neuronal action potential characterized by rapid depolarization due to Na^+^ influx, followed by repolarization by K^+^ efflux, lasting 1–2 ms. (**B**) Longer cardiac action potential (200–300 ms) involving a plateau phase due to sustained Ca^2^^+^ influx via L-type calcium channels, in addition to Na^+^ influx and K^+^ efflux. (**C**) Synaptic transmission in neurons, where neurotransmitter release activates postsynaptic receptors. (**D**) Intercellular transmission in the heart via gap junctions located in intercalated discs, allowing rapid electrical signal conduction between cardiomyocytes.

**Figure 4 biomedicines-13-01376-f004:**
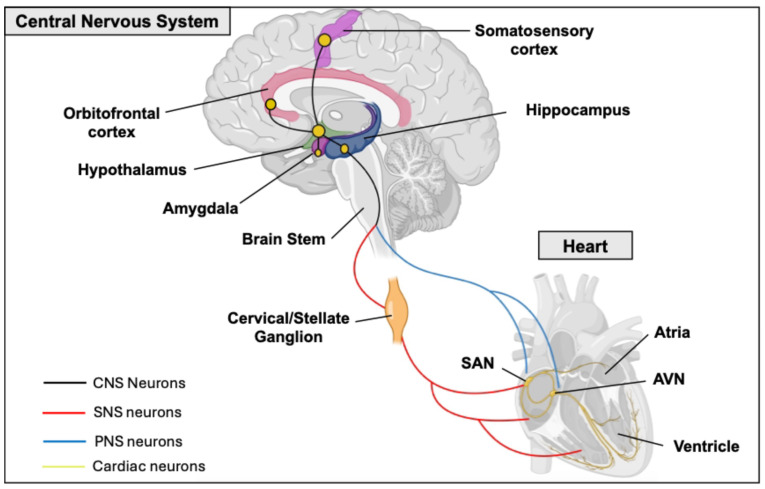
**Schema illustrating the central nervous system (CNS) control of heart function via the autonomic nervous system. Central Nervous System (CNS):** Depicts a sagittal view of the human brain highlighting key regions involved in regulating the heart: **Orbitofrontal Cortex:** Involved in emotional processing and decision-making. **Hypothalamus:** Regulates various bodily functions, including heart rate and blood pressure. **Amygdala:** Processes emotions, particularly fear and stress. **Hippocampus:** Involved in memory formation. **Somatosensory Cortex:** Processes sensory information. **Brain Stem:** Connects the brain to the spinal cord and contains centers that control autonomic functions. **Heart:** Shows a simplified representation of the human heart with its major components: **Atria:** The upper chambers of the heart. **Ventricle:** The lower chamber of the heart. **SAN (Sinoatrial Node):** The heart’s natural pacemaker, initiating the electrical impulses for contraction. **AVN (Atrioventricular Node):** Relays the electrical signal from the atria to the ventricles. **Cervical/Stellate Ganglion:** Represents a key ganglion in the sympathetic nervous system pathway to the heart.

**Table 1 biomedicines-13-01376-t001:** Most relevant Cav1.3 variants implicated in brain and heart diseases.

Variant	Effect	Pathology Associated	Reference
A749G	Gain of function	Parkinson	[35,41]
G407R	Gain of function	Autism	[42]
V401L	Gain of function	Autism	[37]
G1169D	Gain of function	Autism	[36]
R930H	Loss of function Gain of function	Sinus node Dysfunction Epilepsy	[22]
V259D	Gain of function	Sinoatrial dysfunction Deafness (SNND)	[21]
G403D	Gain of function	Sinoatrial dysfunction Deafness (SNND)	
F747L	Gain of function	Sinoatrial dysfunction Deafness (SNND)	
A376V	Loss of function	Sinoatrial dysfunction Deafness (SNND)	[23]
I770M	Shift of activation	Atrioventricular dysfunction	[24]

**Table 2 biomedicines-13-01376-t002:** Localization of Cav1.3 expression in brain and heart.

Organ	Area	Function		References
		Brain	Heart	
Brain	Hippocampus	Neuronal excitability		[43,48]
	Striatum	Neuronal excitability		[41,50]
	Hypothalamus Paraventricular nucleus	Undetermined		[52]
	Amygdala	Neuronal excitability		[38,53,54]
	Parabranchial nucleus	Undetermined		[55]
	Brainstem	Undetermined Auditory function		[56,57]
Heart	Sinoatrial node		Pacemaker function Adrenergic response	[58,59]
	Atria		Action Potential shape	[24]
	Atrioventricular node		Electrical conduction	[60]

## Data Availability

Not applicable.

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
