# Peer review of "Role of Cav1.3 Channels in Brain–Heart Interactions: An Unexpected Journey"

_biomedicines, 2025, doi:10.3390/biomedicines13061376_

Round 1
Reviewer 1 Report
Comments and Suggestions for Authors
This review provides a comprehensive overview of the Cav1.3 L-type calcium channel’s role in brain-heart interactions, integrating findings from molecular physiology, neurocardiology, and genetic disorders. The authors effectively highlight the bidirectional communication between the central nervous system (CNS) and heart, emphasizing Cav1.3’s dual expression in these tissues and its implications for autonomic nervous system (ANS) regulation and disease. The inclusion of cutting-edge technologies like hiPSC-derived neurocardiac models adds translational relevance. However, several areas could be refined to enhance clarity, depth, and critical analysis.
Strengths
- The review systematically covers Cav1.3’s structure, expression patterns in the CNS (hippocampus, striatum, amygdala, etc.) and heart, and its functional roles in neuronal excitability, synaptic plasticity, and cardiac electrophysiology. The integration of genetic variants (e.g., CACNA1D mutations in autism, sinus bradycardia) effectively links basic science to clinical phenotypes.
- The discussion of Cav1.3’s role in action potential dynamics (e.g., pacemaker function in the sinoatrial node) and ANS modulation (e.g., PKA/PKC-mediated channel regulation) is robust. The distinction between Cav1.2 and Cav1.3 functions in different cardiac regions (ventricles vs. atria) clarifies their specialized roles.
- The section on hiPSC models and neurocardiac co-culture systems (e.g., microfluidic devices, optical stimulation) demonstrates awareness of emerging tools for studying brain-heart interactions. This forward-looking approach enhances the review’s relevance to future research.
Areas for Improvement
- The current figures (e.g., Fig. 1 on ANS regulation, Fig. 3 on action potentials) are conceptually valuable but could be enhanced with clearer labeling (e.g., specifying ion channel subtypes in synaptic vs. gap junctional transmission). Consider adding a schematic of Cav1.3’s structural domains (e.g., α1D subunit, accessory proteins) to visually reinforce its molecular architecture.
- While the review highlights Cav1.3’s roles in CNS regions like the hypothalamus and brainstem, it understates the gaps in mechanistic understanding (e.g., unclear how Cav1.3 influences PVN neuronal firing patterns or brainstem-mediated ANS control). Explicitly acknowledging these knowledge gaps would strengthen the review’s analytical rigor.
- The mention of Cav1.3 antagonists for Parkinson’s disease (e.g., levodopa-induced dyskinesia) and arrhythmias is brief. Elaborate on clinical trials or preclinical data evaluating Cav1.3-targeted therapies, including challenges in achieving tissue-specific blockade (e.g., avoiding CNS side effects when targeting cardiac Cav1.3).
- Contrast Cav1.3’s role in brain-heart interactions with other L-type channels (e.g., Cav1.2) more explicitly. For example, while Cav1.2 is dominant in cardiac excitation-contraction coupling, how does Cav1.3’s slower inactivation kinetics (e.g., in SAN cells) uniquely contribute to pacemaker function?
- Some citations are dated (e.g., 2015 for autism spectrum disorder variants). Incorporate recent studies (2023–2025) on Cav1.3’s role in neurodevelopmental disorders or cardiac arrhythmias, such as newly identified CACNA1D variants or mechanistic insights from single-cell sequencing.
- The discussion of the "neurovisceral integration model" (Line 39) could be expanded to clarify how cortical regions (e.g., prefrontal cortex) modulate subcortical ANS centers (e.g., amygdala, hypothalamus) in the context of Cav1.3 expression. This would strengthen the "brain-heart axis" narrative.
This review offers a valuable synthesis of Cav1.3’s contributions to brain-heart crosstalk, suitable for readers interested in ion channel physiology, neurocardiology, and translational medicine. With refinements to figure clarity, critical discourse on limitations, and expanded therapeutic/technological insights, the manuscript will become even more impactful. The authors are commended for their thoroughness, and the suggested revisions will enhance the review’s depth and utility to the field.
Author Response
We sincerely thank the reviewers for their time and for the constructive comments. We have addressed each and every comment below and indicated the corresponding changes in red in the revised manuscript. As such the revised manuscript has been significantly improved.
Comment 1:
- The current figures (e.g., Fig. 1 on ANS regulation, Fig. 3 on action potentials) are conceptually valuable but could be enhanced with clearer labeling (e.g., specifying ion channel subtypes in synaptic vs. gap junctional transmission)Consider adding a schematic of Cav1.3’s structural domains (e.g., α1D subunit, accessory proteins) to visually reinforce its molecular architecture.
In Figure 2, we've added section B to depict the assembly of the pore-forming LTCC Cav1.3 and its regulatory subunit complex. Furthermore, Figure 3 has been expanded to incorporate the most significant ion channel subtypes influencing action potential shape in both neurons and cardiomyocytes, and to show the ion channel and connexin subunits at the presynaptic terminal of synapses and gap junctions respectively.
Comment 2:
2. While the review highlights Cav1.3’s roles in CNS regions like the hypothalamus and brainstem, it understates the gaps in mechanistic understanding (e.g., unclear how Cav1.3 influences PVN neuronal firing patterns or brainstem-mediated ANS control). Explicitly acknowledging these knowledge gaps would strengthen the review’s analytical rigor.
The function of the Cav1.3 calcium channel has been well-characterized in most brain regions; however, its specific role in regulating neuronal firing patterns within the PVN remains poorly understood, with clear studies currently lacking. We can only speculate that, analogous to other areas, including the suprachiasmatic nucleus (another hypothalamic region), the Cav1.3 channel mediates calcium influx, which subsequently activates the calcium-activated potassium (BK) channel, thus supporting a normal firing pattern. (McNally et al 2021 DOI:10.3389/fphys.2021.737291
In lines 281 to 284, we clarified that the brainstem, with the medulla oblongata being a key component, contains central autonomic control nuclei. These nuclei integrate input from various brain and spinal cord areas before sending signals to the peripheral ANS to regulate cardiac function. (Martin-Gallego et al., 2017 DOI: 10.1007/978-3-319-39546-3_34
Comment 3:
3. The mention of Cav1.3 antagonists for Parkinson’s disease (e.g., levodopa-induced dyskinesia) and arrhythmias is brief. Elaborate on clinical trials or preclinical data evaluating Cav1.3-targeted therapies, including challenges in achieving tissue-specific blockade (e.g., avoiding CNS side effects when targeting cardiac Cav1.3).
Medications intended for the treatment of neurological conditions tend to more readily induce adverse effects in other organ systems (as exemplified by the review showing levodopa-induced arrhythmias in certain patients). The converse is observed less frequently due to the presence of the blood-brain barrier. The limited availability of medications specifically targeting Cav1.3 currently prevents us from ascertaining whether such targeting could lead to adverse effects in other areas, notably the brain.
Comment 4:
4. Contrast Cav1.3’s role in brain-heart interactions with other L-type channels (e.g., Cav1.2) more explicitly. For example, while Cav1.2 is dominant in cardiac excitation-contraction coupling, how does Cav1.3’s slower inactivation kinetics (e.g., in SAN cells) uniquely contribute to pacemaker function?
Cav1.3 exhibits distinct biophysical properties, such as slower inactivation kinetics and activation at more hyperpolarized potentials compared to Cav1.2. These characteristics are particularly significant in the sinoatrial node (SAN) cells. The slower inactivation of Cav1.3 allows for a sustained calcium influx during the later stages of the pacemaker potential (diastolic depolarization). While Cav1.2 is also present in SAN cells, its faster inactivation kinetics make its contribution to the later phases of diastolic depolarization less prominent compared to Cav1.3. This clarification was added Line 368 to 375.
Comment 5:
5. Some citations are dated (e.g., 2015 for autism spectrum disorder variants). Incorporate recent studies (2023–2025) on Cav1.3’s role in neurodevelopmental disorders or cardiac arrhythmias, such as newly identified CACNA1D variants or mechanistic insights from single-cell sequencing.
Reference added as much as we could
Comment 6:
6. The discussion of the "neurovisceral integration model" (Line 39) could be expanded to clarify how cortical regions (e.g., prefrontal cortex) modulate subcortical ANS centers (e.g., amygdala, hypothalamus) in the context of Cav1.3 expression. This would strengthen the "brain-heart axis" narrative.
We have added some details about the neurovisceral integration model Line 73 to 77: The cortex especially plays a crucial role in modulating the limbic system, acting as a top-down control mechanism to regulate emotions, memories and behaviors (14,15). The cortex interacts with limbic structures through various neurotransmitter such as glutamate. The cortex allow an appropriate and adaptative behavioral responses to various situations (16).
Reviewer 2 Report
Comments and Suggestions for Authors
Very nice and well-written review on which the function of Cav1.3 L-type calcium channel in brain-heart interaction is discussed. The review is illustrated with important figures and tables, however there are still some questions that could improve this review.
- I will recommend to the authors to include in the review also information concerning Cav1.2 especially because it is mentioned that “In the heart and brain Cav1.2 is the predominant L-type Ca2+ channel, while Cav1.3 is generally less abundant” Both this channels play important role in regulation of brain and heart calcium hemostasis and including Cav1.2 will make this review much more interesting. Some information concerning Cav1.2 is already presented, however will be better if this information will be expanded and included in the title.
- In the table 2 better to divide function for brain and heart.
- In my opinion the chapter “5. Neuromuscular experimental modalities and developments” is too long and contain many information which have no direct connections with the presented review.
- 9, L 293-4. “Specifically, two forms of Cav1.3 have been identified: a full-length 250 kD protein, which is the predominant form prenatally, and a shorter 190 kD form.” Which part of the short Cav1.3 is missing? Is anything known regarding functional differences and tissue distribution of these two isoforms? In the figure 2 will be good to include the structure of the short Cav1.3.
Minor.
- 2, L 38. “strong increase in cAMP and PKA” Can not be increase of PKA, PKA activity.
- 4, L. 97. “ (Nerbonne et Kass 2005).” Should be corrected.
- 4, L. 112. “a distinctly different APs morphology”. I do not think that morphology is correct to characterize AP.
- 6, L. 168. “They showed that the loss of Cav1.3 IQ-domain RNA editing is a gain of function and increased Cav1.3 channel current and led to enhanced neuronal excitability compared to Cav1.3 editing.” May be compared to control? The sentence should be revised.
- 8, L. 228. “the heart:Orbitrofrontal”. Space is missing.
- 10, L. 305. “the induction of calcium induced calcium release.” Release from cells or ER? This should be corrected.
Author Response
We sincerely thank the reviewers for their time and for the constructive comments. We have addressed each and every comment below and indicated the corresponding changes in red in the revised manuscript. As such the revised manuscript has been significantly improved.
Comment 1:
- I will recommend to the authors to include in the review also information concerning Cav1.2 especially because it is mentioned that “In the heart and brain Cav1.2 is the predominant L-type Ca2+ channel, while Cav1.3 is generally less abundant” Both this channels play important role in regulation of brain and heart calcium hemostasis and including Cav1.2 will make this review much more interesting. Some information concerning Cav1.2 is already presented, however will be better if this information will be expanded and included in the title.
We have added some clarifications of Cav1.2 in heart, especially the role in SAN cells and the difference with Cav1.3 Line 368 to 375. We also added few information in the introduction part line 85 to 91, concerning Cav1.2 and cav1.3.
We acknowledge your point concerning Cav1.2 in the brain and heart; nevertheless, the scope of this review will be centered on Cav1.3. We hypothesize that Cav1.3 is an underappreciated player in the brain and the brain-heart axis, largely due to its distinctive biophysical and inactivation kinetic characteristics that offer compelling opportunities for future research.
Comment 2:
2. In the table 2 better to divide function for brain and heart.
We separated the column Function in 2 (Brain and Heart) for more clarity
Comment 3:
3. In my opinion the chapter “5. Neuromuscular experimental modalities and developments” is too long and contain many information which have no direct connections with the presented review.
Your comment regarding the length of this section was taken into consideration, and we did try to condense it. However, the other reviewer found the section valuable and requested its retention, resulting in only minimal changes.
Comment 4:
4. 9, L 293-4. “Specifically, two forms of Cav1.3 have been identified: a full-length 250 kD protein, which is the predominant form prenatally, and a shorter 190 kD form.” Which part of the short Cav1.3 is missing? Is anything known regarding functional differences and tissue distribution of these two isoforms? In the figure 2 will be good to include the structure of the short Cav1.3.
The long Cav1.3 isoform dominates the prenatal stage, while a C-terminally truncated short isoform is specific to the adult atria. Moreover, Cav1.3 undergoes significant alternative splicing in the CNS and heart, a characteristic facilitated by its long C-terminal domain. For example, the full-length Cav1.3 isoform (Cav1.342L) contains all regulatory domains. In contrast, two shorter splice variants, Cav1.342A and Cav1.343S, lack the distal C-terminal regulatory domain or both the proximal and distal domains (Stanika et al., 2016). Alternative splicing in the Cav1.3 C-terminus is known to alter its electrophysiological properties (Hofer et al., 2021). For instance, Cav1.342Achannels exhibit ICaL activation at more negative voltages and faster inactivation due to enhanced Ca2+-dependent inactivation (Singh et al., 2008). Moreover, the C-terminal modulator domain in Cav1.342 isoforms can compete with calmodulin (CaM) for binding to the IQ domain (Kuzmenkina et al., 2019). This precision was added line 326 to 335.
Round 2
Reviewer 2 Report
Comments and Suggestions for Authors
The authors adequately addressed all points and I have no more questions.